# A Comprehensive Framework for Benchmarking Algorithms Across Hyperparameter Spaces

## Abstract

We introduce a framework for benchmarking algorithms with varying hyperparameters from multiple perspectives. The dependency of algorithms' performance on hyperparameters complicates fair comparisons and often leads to inconsistent empirical studies. Our framework addresses this challenge by proposing two key criteria: *Performance-HPO* trajectory and *Reliability-HPO*. The Performance-HPO trajectory tracks how an algorithm's performance changes with different hyperparameter optimization (HPO) budget allocations, leveraging a variety of off-the-shelf hyperparameter optimizers. This enables users to identify the most suitable algorithm for their specific needs. The Reliability-HPO criterion evaluates the expected value of an algorithm's success rate across hyperparameters, estimated using Monte Carlo simulations in log-space. We demonstrate our framework by benchmarking widely-used convex optimizers. Our experiments, conducted with `CVXPY` across various problem types, settings, and dimensionalities, reveal that the `SCS` solver exhibits the highest Performance-HPO, while `ECOS` and `MOSEK` demonstrate superior Reliability-HPO.

## 1 Introduction

Most algorithms require hyperparameter optimization to achieve optimal performance. Convex optimizers such as `MOSEK` (ApS, 2024), `SCS` (O'Donoghue et al., 2016; O'Donoghue, 2021), and `OSQP` (Ichnowski et al., 2021; Stellato et al., 2020) each have different hyperparameters. Optimizing these hyperparameters impacts their performance significantly (Ichnowski et al., 2021). Despite this, no straightforward method exists to optimize many of these hyperparameters in advance. For instance, selecting optimal parameters for the Alternating Direction Method of Multipliers (ADMM) remains an open research question (Ghadimi et al., 2014; Giselsson & Boyd, 2016; Nishihara et al., 2015). To address this, Ichnowski et al. (2021) propose using reinforcement learning to tune `OSQP` hyperparameters.

In other fields, this issue also affects performance. In machine learning, hyperparameter choice often causes inconsistent results in empirical work (Cooper et al., 2021; Gundersen et al., 2022). Approximately 45% of hyperparameter optimization in NeurIPS 2019 and ICLR 2020 papers is manual (Gundersen et al., 2022). Researchers with a lot of computing resources are able to fine-tune the of an algorithm's hyperparameters more than others. This makes it hard to compare the effectiveness of different algorithms. Probst et al. (2019) discuss the importance of machine learning algorithm hyperparameters. Similarly, Sivaprasad et al. (2020) explore the profound impact of hyperparameter optimization on deep learning optimizers' performance. Moreover, Cooper et al. (2021) emphasize hyperparameter optimization's significance in machine learning. They suggest that deriving meaningful insights from this process warrants dedicated research. However, the literature still needs an unbiased methodology to benchmark different algorithms effectively.

In machine learning, hyperparameter optimization is a stochastic bilevel optimization problem (Beck et al., 2023). A training set tunes model parameters, and a validation set tunes model hyperparam-

eters. Optimal parameters are a function of hyperparameters. Problem 1 shows hyperparameter optimization formulated as a stochastic bilevel optimization problem.

$$\min_{x \in \mathcal{X}, \; y^*(x) \in \arg\min_{y \in \mathcal{Y}} g(x,y)} \; \mathbb{E}_{\xi \sim \Xi} \; f\big(x, y^*(x); \xi\big), \tag{1}$$

where $y$ indicates model parameters, and $x$ denotes model hyperparameters. $f$ and $g$ are the validation and training losses, respectively. $\xi$ is a random variable, and $y^*(x)$ indicates optimal parameters given hyperparameters $x$. This problem is challenging because the optimal weights $y^*(x)$ are a function of the hyperparameters $x$. Hence, in many empirical studies, researchers often do not tune hyperparameters reasonably. This leads to inconsistent results.

We provide an example to clarify the problem and the intuition behind our methodology. Figure 1 shows the performance of two hypothetical algorithms, A and B, given the hyperparameter optimization budget. Algorithm B performs better with low hyperparameter optimization budgets. It is less sensitive to its hyperparameters. Algorithm A requires a substantial hyperparameter optimization budget to outperform Algorithm B. Researchers repeatedly solve problems with different data in applications such as model predictive control. In such applications, spending extra time optimizing Algorithm A's hyperparameters pays off in the total performance across all problem instances.

Algorithms A and B are suitable for specific use cases. It depends on the hyperparameter optimization budget. We call the curves in Figure 1 the *Performance-HPO* trajectories of Algorithms A and B. We propose to use it to compare their performance. Given the Performance-HPO trajectory, the user decides which algorithm to use. This decision depends on the settings. We define Performance-HPO trajectory in Section 3 formally. This methodology is trivially generalizable to settings with more than two algorithms. Additionally, we are able to calculate the area under the curve to measure the overall performance of each algorithm across a range of hyperparameter optimization budgets.

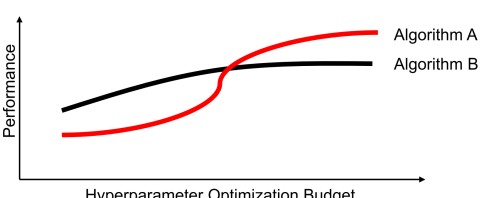

Figure 1: Performance of two hypothetical algorithms based on hyperparameter optimization budget. In near-optimal settings, Algorithm A outperforms Algorithm B. However, it requires a substantial hyperparameter optimization budget. Without hyperparameter optimization, Algorithm B outperforms Algorithm A. In this example, it is not trivial which algorithm is superior.

To use this methodology, we define an unbiased method to obtain the Performance-HPO trajectory for each algorithm. Previous work utilizes only a single hyperparameter optimizer, such as random search (Sivaprasad et al., 2020). This approach ignores various off-the-shelf hyperparameter optimizers such as Bayesian optimization (Garnett, 2023) and evolutionary algorithms (Bäck & Schwefel, 1993). Practitioners widely use these optimizers in real-world applications (Nogueira, 2014–; Gad, 2023). Some packages provide a unified interface for utilizing multiple hyperparameter optimizers. Examples include `nlopt` (Johnson, 2011), `gradient-free-optimizers` (Simon Blanke, 2020), `hyperactive` (Simon Blanke, since 2019), `hyperopt` (Bergstra et al., 2013), and `scikit-optimize` (Head et al., 2018).

To address the limitations of the past benchmarking methods, we suggest simultaneously using a list of off-the-shelf hyperparameter optimization algorithms. This approach aims to calculate the Performance-HPO trajectory. Each hyperparameter optimization algorithm improves algorithm performance by optimizing its hyperparameters separately. At each time step, we select the best result from the hyperparameter optimization algorithms to define the Performance-HPO trajectory at that time.

Reliability based on hyperparameter choice is another important criterion when comparing algorithms with different hyperparameters. Consider two iterative algorithms whose success depends on convergence. It is essential to understand how difficult it is to find hyperparameters that lead to convergence. The definition of "success" for an algorithm varies by area. For example, for convex optimizers, success means finding a solution that meets pre-defined accuracy metrics within a specified time.

We define the *Reliability-HPO* of an algorithm as the expected value of its success given random hyperparameters. We propose using Reliability-HPO to evaluate the reliability of different algorithms. Algorithms without hyperparameters highlight the importance of computing Reliability-HPO in addition to Performance-HPO trajectory. For these algorithms, the Performance-HPO trajectory is flat since there are no hyperparameters to optimize for improved performance. Thus, algorithms with hyperparameters potentially outperform those without hyperparameters under high optimization budgets. However, their Reliability-HPO will be at the maximum value. This highlights their reliability. This example demonstrates why considering both performance and reliability based on hyperparameters provides a complete framework for benchmarking algorithms with different hyperparameters.

We demonstrate our methodology using convex optimizers. Practitioners widely use convex optimization in fields such as finance (Boyd et al., 2017), signal processing (Luo, 2003), and modeling uncertainty (Ben-Haim & Elishakoff, 2013). One challenge with convex optimizers is formatting the optimization problem in the specific form required by the optimizer. CVXPY (Diamond et al., 2014) simplifies this by converting problems from a user-friendly format to the required optimizer's format. Given its extensive use, we use CVXPY to benchmark convex optimizers. The SCS solver achieves the highest Performance-HPO. In contrast, ECOS and MOSEK exhibit the highest Reliability-HPO.

**Contributions.** The contributions of this paper are as follows:

- Methodologically, we introduce the Performance-HPO trajectory and Reliability-HPO. These criteria provide a generic framework to measure the performance and reliability of algorithms based on their hyperparameters. Our methodology is unbiased and does not involve human intervention. It applies to any algorithm, regardless of the domain.
- Empirically, we use our framework to benchmark convex optimizers through CVXPY. We perform extensive real-world experiments across different problem types, settings, and dimensionalities.

The structure of the rest of the paper is as follows: we provide the details of the related work and the proposed methodology in Section 2 and Section 3, respectively. We discuss the experiments in Section 4. Section 5 concludes the paper.

## 2  RELATED WORK

**Impact of Hyperparameter Selection on Reproducibility.** Melis et al. (2017) re-evaluate state-of-the-art architectures and regularization techniques in neural language models. They conclude if one sufficiently optimizes hyperparameters of standard LSTM architectures, they outperform newer models. Reimers & Gurevych (2017) explore why LSTM networks show different performance levels in sequence tagging tasks. They focus on how different hyperparameter optimization methods contribute to these differences. They compare random search, grid search, and Bayesian optimization. They find that the results significantly vary depending on the hyperparameter optimizer. Lucic et al. (2018) benchmark generative adversarial networks. They conclude that many models perform similarly with sufficient hyperparameter optimization budget. Dodge et al. (2019) study the effect of hyperparameters in natural language processing. They show it is impossible to rank models solely based on their scores on the test set. They define expected validation performance as a function of the number of hyperparameter trials. Choi et al. (2019) show optimizer comparisons are sensitive to the hyperparameter tuning protocol. They argue that the hyperparameter search space might be the most crucial factor. It explains the rankings obtained by recent empirical comparisons in that area. They demonstrate that changing the hyperparameter search spaces potentially leads to contradictory results. Ferrari Dacrema et al. (2021) analyze the reproducibility of recommender systems papers in reputable conferences. They discover that simple baselines potentially outperform 11 of the 12 reproducible neural approaches from 2015 to 2018. This is because the researchers do not adequately optimize the baselines in their papers. Gundersen et al. (2022) review sources of irreproducibility in machine learning. They identify differences in computational budgets and selective tuning of algorithms as the leading causes of irreproducibility. They suggest that researchers should specify the exact methods used for hyperparameter optimization.

**Guidelines for Incorporating Hyperparameters in Benchmarking.** Lucic et al. (2018) propose reporting the distribution of Fréchet Inception Distance (FID) scores in generative adversarial networks. They state that researchers should look at the entire range of scores, not just the minimum. Sivaprasad et al. (2020) argue that benchmarking deep learning optimizers must consider hyperparameter optimization. Their study focuses on two aspects: (I) the best performance achieved by

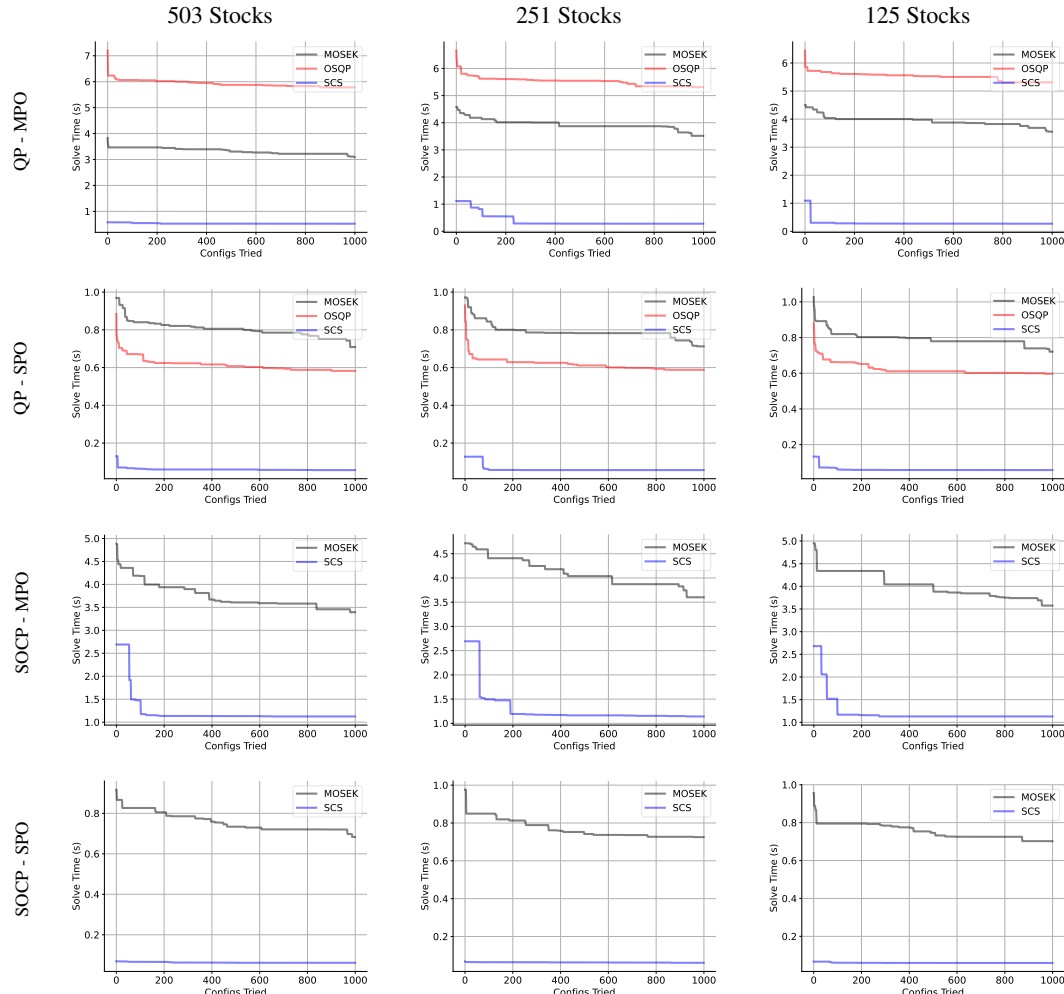

Figure 2: Performance-HPO trajectories of the convex optimizers across problem types, settings, and dimensionalities. We use the following acronyms: QP (Quadratic Programming), SOCP (Second-Order Cone Programming), SPO (Single-Period Optimization), and MPO (Multi-Period Optimization).

an optimizer and (II) the difficulty in finding the hyperparameters that lead to that performance. They use random search as the hyperparameter optimization method for their benchmark. Xiong et al. (2020) argue random search over-emphasizes the tuning time and is expensive. As a result, evaluations based on this method often focus on the low-accuracy region. However, one is more concerned with achieving reasonably good accuracy in practical applications. They propose to use Hyperband (Li et al., 2018). They state that Hyperband reflects the real hyperparameter optimization time required for the user more accurately. Cooper et al. (2021) argue that the methodology used for deriving insights through hyperparameter optimization should itself be studied. They refer to this process as Epistemic Hyperparameter Optimization (EHPO) and introduce a logical framework to describe its meaning. This framework also demonstrates how EHPO is able to result in inconsistent conclusions about performance. This is the first characterization supported by theory. They design it to make reliable conclusions about algorithm performance using hyperparameter optimization.

## 3 PROPOSED METHODOLOGY

This section introduces a framework to objectively benchmark different algorithms across different problems. Our criteria include Performance-HPO and Reliability-HPO. Consider a collection of problems denoted as $r_1, \ldots, r_{n_r}$. $n_r$ is the number of benchmarking problems. There are a set of algorithms labeled $g_1, \ldots, g_{n_g}$. $n_g$ denotes the number of algorithms. Each algorithm $g_i$ is

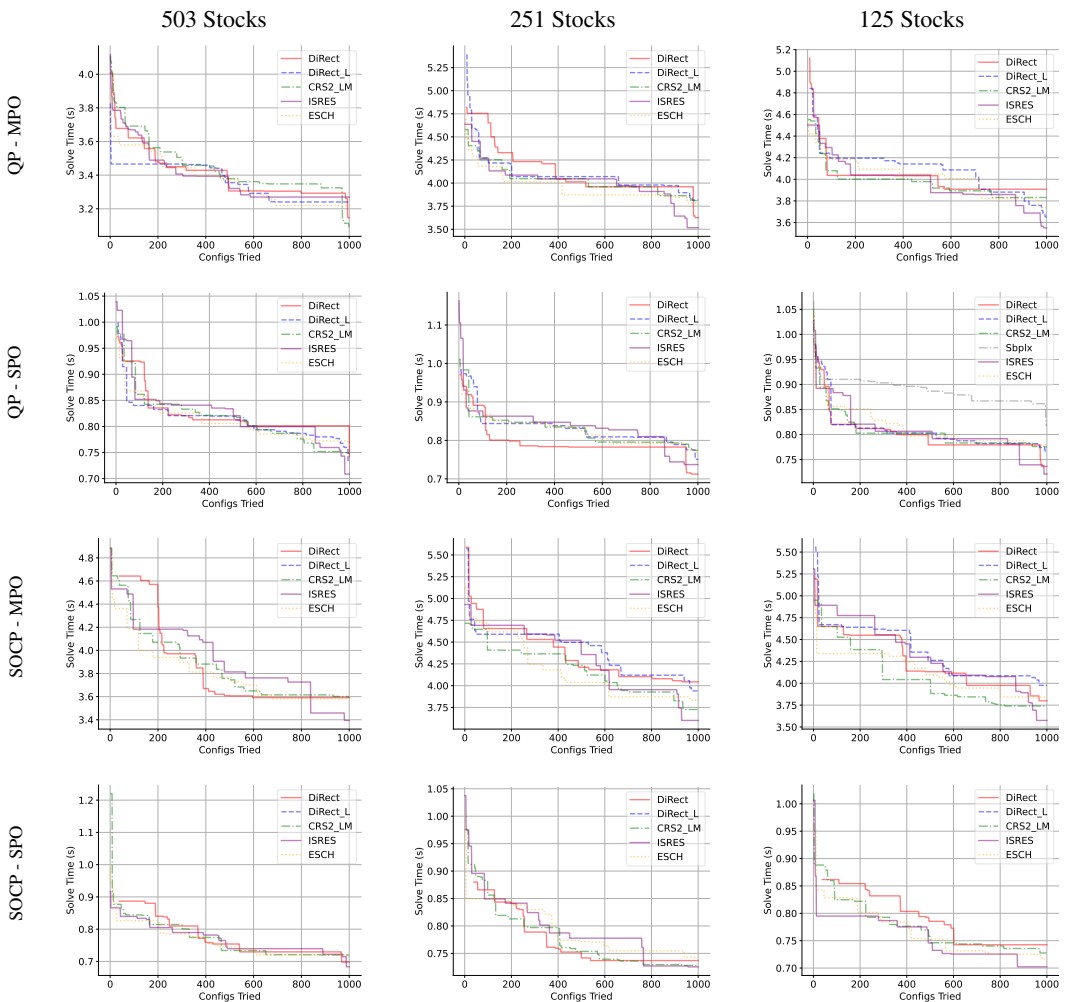

Figure 3: Hyperparameter optimization for MOSEK across problem types, settings, and dimensionalities. We use the following acronyms: QP (Quadratic Programming), SOCP (Second-Order Cone Programming), SPO (Single-Period Optimization), and MPO (Multi-Period Optimization).

applicable to each problem $r_j$, where $i \in \{1 \cdots n_g\}$ and $j \in \{1 \cdots n_r\}$. Each algorithm $g_i$ has $m_i$ hyperparameters, where $i \in \{1 \cdots n_g\}$.

**Performance-HPO.** This criteria evaluates an algorithm's performance based on the hyperparameter optimization budget. We use a series of hyperparameter optimizers to optimize these algorithms. We represent the hyperparameter optimizers as $h_1, \ldots, h_{n_h}$. $n_h$ is the number of hyperparameter optimizers. Given hyperparameter optimization budget $t$, a hyperparameter optimizer $h_j$ optimizes hyperparameters of an algorithm $g_k$ for problem $r_i$. We denote the algorithm with the optimized hyperparameters with $g_k^{(h_j, r_i, t)}$. Then, we evaluate the performance of the optimized algorithm on that problem with $p\left(g_k^{(h_j, r_i, t)}, r_i\right)$, where $p$ denotes the performance metric. We run all hyperparameter optimizers in parallel and select the hyperparameter optimizer with the highest performance. Problem 2 provides the formulation.

$$\max_{h_j \in \{h_1, \ldots, h_{n_h}\}} p\left(g_k^{(h_j, r_i, t)}, r_i\right) \tag{2}$$

We create the Performance-HPO trajectory by integrating the solution to Problem 2 for $0 \leq t \leq T$. $t$ denotes the hyperparameter optimization budget. We provide the formulation for the Performance-HPO trajectory in Problem 3.

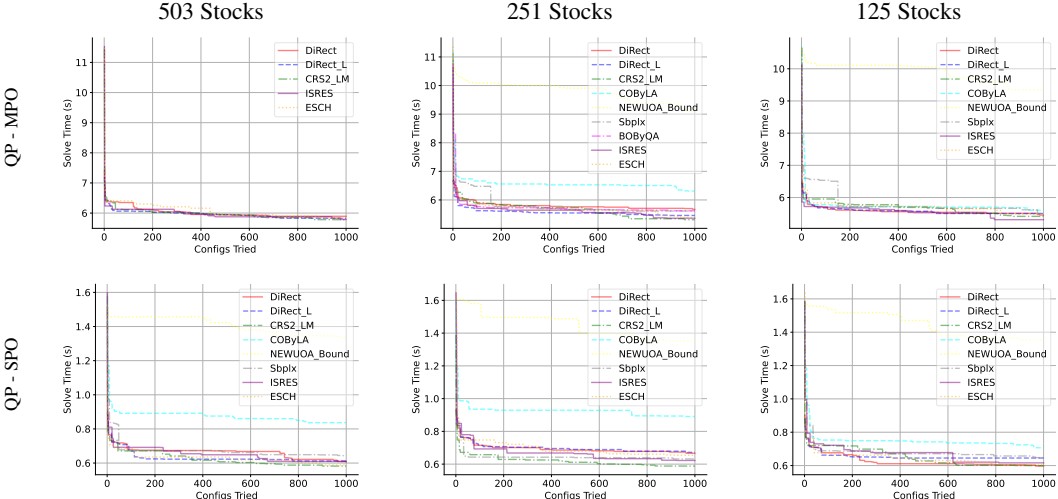

Figure 4: Hyperparameter optimization for `OSQP` across problem types, settings, and dimensionalities. We use the following acronyms: QP (Quadratic Programming), SOCP (Second-Order Cone Programming), SPO (Single-Period Optimization), and MPO (Multi-Period Optimization).

$$\max_{h_j \in \{h_1, \dots, h_{n_h}\}} \left( p \left( g_k^{(h_j, r_i, t)}, r_i \right) \right), \quad t \in [0, T] \tag{3}$$

We propose to use a spectrum of off-the-shelf hyperparameter optimization methods. Practitioners widely use hyperparameter optimizers such as random search (Liaw et al., 2018), Bayesian optimization (Nogueira, 2014–), and evolutionary algorithms (Gad, 2023) in various real-world applications. We believe this methodology simulates how real-world applications utilize algorithms. Several off-the-shelf packages provide a unified interface for hyperparameter optimizers (Johnson, 2011; Simon Blanke, 2020; since 2019; Bergstra et al., 2013; Head et al., 2018). They facilitate running hyperparameter optimizers in parallel.

**Reliability-HPO.** For an algorithm, it is important to measure how difficult it is to find hyperparameters that lead to a successful run. *success* depends on the algorithm's context. For an iterative algorithm, the run is successful if it achieves a pre-specified accuracy within a given time. This measure of reliability is crucial. If the algorithm fails, one needs to rerun it with new hyperparameters. The success of the new hyperparameters is not guaranteed either.

We define Reliability-HPO as the expected success rate of an algorithm on a problem. We estimate this expectation using Monte Carlo simulation. For this purpose, we sample a large number of hyperparameters from log-uniform distribution. Then, we compute the average success rate.

Consider two hypothetical algorithms: A and B. Algorithm A has no hyperparameters. As a result, its Performance-HPO trajectory is constant. If this algorithm successfully solves the problem of interest, its Reliability-HPO is one. Algorithm B, on the other hand, has many hyperparameters. It is able to outperform Algorithm A with a high hyperparameter optimization budget. However, the many hyperparameters make successful runs more challenging. As a result, its Reliability-HPO is lower than Algorithm A. This demonstrates that the Reliability-HPO criterion is necessary to provide a comprehensive benchmark. We provide examples of such algorithms in Section 4.

## 4 EXPERIMENTS

We utilize the proposed framework to benchmark convex optimizers. Convex optimization has numerous applications in portfolio optimization (Boyd et al., 2017), signal processing (Luo, 2003), and aerospace engineering (Liu et al., 2017). We refer to Ben-Tal & Nemirovski (2001) for a comprehensive list of applications. Tuning hyperparameters for most convex optimizers is necessary for optimal performance (Ichnowski et al., 2021). Ghadimi et al. (2014); Giselsson & Boyd (2016); Nishihara et al. (2015) indicate optimizing these hyperparameters is non-trivial. It requires dedicated research (Ichnowski et al., 2021). Despite this, previous benchmarks on convex optimizers ignored the hyperparameters (Kozma et al., 2015).

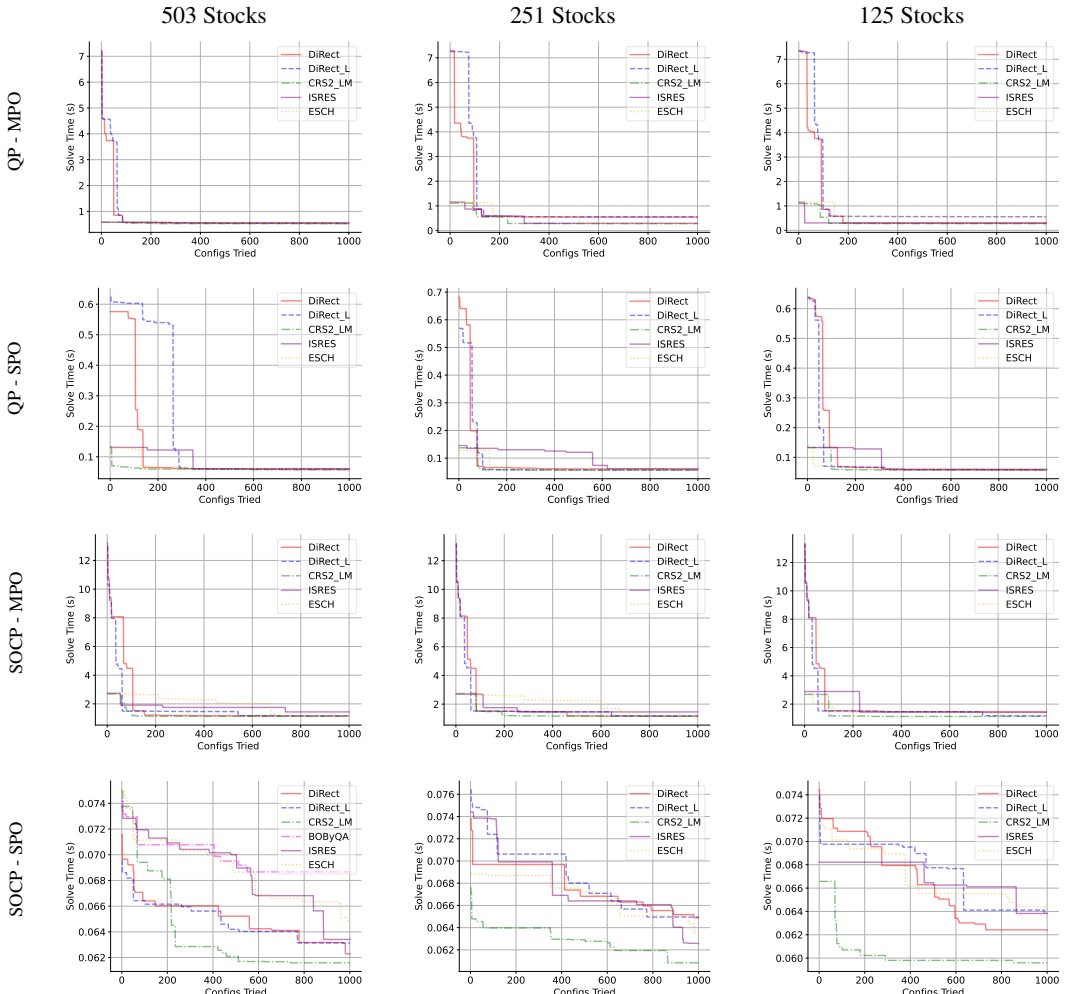

Figure 5: Hyperparameter optimization for SCS across problem types, settings, and dimensionalities. We use the following acronyms: QP (Quadratic Programming), SOCP (Second-Order Cone Programming), SPO (Single-Period Optimization), and MPO (Multi-Period Optimization).

To use a convex optimizer, the problem must fit its required format. Hence, the users need to reformulate it for that solver. This is challenging for non-expert users. Moreover, the required format varies among solvers. Domain-Specific Languages (DSLs) help bridge this gap. CVXPY is a DSL designed for this purpose. It converts a problem from a user-friendly format to a solver-friendly standard form. Users only need to learn CVXPY's format instead of each solver's. Practitioners widely use CVXPY in real-world applications. We use it to benchmark convex optimizers.

**Problem Design.** We design two classes of optimization problems using the CVXPY library: quadratic programming and second-order cone programming. For each problem class, we address two settings: single-period optimization and multi-period optimization. We employ the cvxportfolio package (Busseti et al., 2017) to create real-world optimization problems. We source data for 503 stocks from yfinance (Aroussi, 2020). We investigate solvers' performance across different problem sizes. For this purpose, we vary the stock size by 100%, 50%, and 25%. The designed problems pertain to the stochastic model predictive control framework. Problem 4 provides their formulation.

$$
u_{t|t}^*, \dots, u_{t+H-1|t}^* \in \underset{u_{t|t}, \dots, u_{t+H-1|t}}{\arg\min} \quad \sum_{\tau=t}^{t+H-1} \underset{\xi_\tau \sim \Xi_\tau}{\mathbb{E}} \left[ q \left( u_{\tau|t}, \hat{x}_{\tau|t}; \xi_\tau \right) \right]
$$
$$
\text{subject to} \quad \hat{x}_{\tau+1|t} = \hat{f} \left( u_{\tau|t}, \hat{x}_{\tau|t}; \xi_\tau \right),
$$
$$
u_{\tau|t} \in U \left( \xi_\tau \right).
$$
(4)

where $H$ is the planning horizon. We use three and one for multi-period and single-period optimization, respectively. The multi-period optimization involves three times the optimization variables compared to the single-period optimization. $\hat{f}$ denotes the simplified model dynamics. It is a linear function in our case. $U$ denotes the feasible set for the optimization variables. It corresponds to self-finance, leverage limit, turnover limit, and long-only constraints. Self-finance is an equality constraint. The other constraints are inequality constraints. $q$ denotes the objective function. It consists of a linear term for the expected return. There is also a quadratic term for the risk measure. The holding cost is the absolute value of a linear term. Additionally, there is a transaction cost. In quadratic programming, we use a quadratic estimation of the transaction cost. In second-order cone programming, we utilize the square root of the cube. We refer to Boyd et al. (2017) for more details on the optimization problem.

**Convex Optimizers.** There exist more than 10 convex optimizers through CVXPY: MOSEK (ApS, 2024), OSQP (Stellato et al., 2020; Banjac et al., 2019), SCS (O'Donoghue, 2021; O'Donoghue et al., 2016), ECOS (Domahidi et al., 2013), CVXOPT (Vandenberghe, 2010), SCIP (Achterberg, 2009), Gurobi (Gurobi Optimization, LLC, 2024), CPLEX (IBM ILOG CPLEX Optimization Studio, 2024), Xpress (FICO, 2024), NAG (The Numerical Algorithms Group, 2024), PDLP (Applegate et al., 2021), and CBC (COIN-OR, 2024). Among these solvers, only MOSEK, OSQP, SCS, and ECOS apply to our problem and manage to solve it. Note that OSQP does not apply to second-order cone programming problems. We discuss these solvers in more detail in Appendix A

**Optimizers' Hyperparameters.** We briefly outline the hyperparameters we optimize for each solver. Detailed discussions of these hyperparameters appear in Appendix B. For MOSEK, we focus on optimizing the following hyperparameters: Msk_Dpar_Intpnt_Tol_Rel_Step, Msk_Dpar_Intpnt_Tol_Psafe, Msk_Dpar_Intpnt_Tol_Dsafe, and Msk_Dpar_Intpnt_Tol_Path. For OSQP, the optimized hyperparameters include Rho, Alpha, and Sigma. In the case of SCS, we consider Alpha, Rho_x, Scale, Acceleration_Lookback, and Acceleration_Interval. ECOS does not include any hyperparameters.

## 4.1 PERFORMANCE-HPO

**Hyperparameter Optimizers.** We use off-the-shelf hyperparameter optimizers through the nlopt package. We discuss them in more detail in Appendix C.

**Performance Evaluation.** We report the solve time of each solver as the performance metric. We ensure the solvers reach the same accuracy. This makes the performance evaluation consistent. Problem 5 provides a general formulation for constrained optimization. $\mathbf{0}$ denotes the zero vector. We necessitate accuracy criteria described as follows: $f(x) < \epsilon_0$, $G(x) < \epsilon_1 \mathbf{1}$, and $H(x) < \epsilon_2 \mathbf{1}$, where $\mathbf{1}$ is the unit vector. We set $\epsilon_0$ based on the optimization problem. We assign $\epsilon_1 = \epsilon_2 = 1^{-5}$. Tolerance parameters specify an optimizer's accuracy. Typically, they include relative and absolute tolerance. For each optimizer on each problem, we find the maximum tolerance values satisfying the accuracy criteria. We conduct this using exponential search.

$$\min_x \quad f(x) \tag{5}$$
$$\text{subject to} \quad G(x) \leq \mathbf{0}$$
$$H(x) = \mathbf{0}$$

**Results.** Figure 2 shows the Performance-HPO trajectories of MOSEK, OSQP, and SCS across problem types, settings, and dimensionalities. We provide the details of the hyperparameter optimizations in Figures 3 to 5. ECOS requires much more time to solve the optimization problems. Moreover, its Performance-HPO trajectory is a con-

Table 1: Performance of ECOS across problem types, settings, and dimensionalities. We provide mean and standard deviation over five runs. ECOS has no hyperparameters. Hence, its Performance-HPO trajectory is a constant. We use the following acronyms: QP (Quadratic Programming), SOCP (Second-Order Cone Programming), SPO (Single-Period Optimization), and MPO (Multi-Period Optimization).

|  | 503 Stocks | 251 Stocks | 125 Stocks |
|---|---|---|---|
| QP - SPO | $4.54 \pm 0.05$ (s) | $4.50 \pm 0.08$ (s) | $4.50 \pm 0.03$ (s) |
| QP - MPO | $40.63 \pm 3.8$ (s) | $40.27 \pm 6.7$ (s) | $40.99 \pm 5.9$ (s) |
| SOCP - SPO | $3.62 \pm 0.06$ (s) | $3.63 \pm 0.07$ (s) | $3.51 \pm 0.09$ (s) |
| SOCP - MPO | $41.99 \pm 4.4$ (s) | $40.92 \pm 2.6$ (s) | $41.10 \pm 8.1$ (s) |

stant. Hence, we provide its results in Table 1 instead of the figures. In quadratic programming, SCS dominantly outperforms MOSEK and OSQP regardless of the hyperparameter optimization budget. OSQP applies only to quadratic programming. But MOSEK outperforms it in this problem type.

In second-order cone programming, SCS outperforms MOSEK regardless of the hyperparameter optimization budget and problem dimensionality. In single-period optimization, number of variables varies from 125 to 503. In multi-period optimization, the planning horizon is three. Hence, the number of variables ranges from 375 to 1509. The problem dimensionality in these ranges does not significantly affect optimizers' perfor-

Table 2: Reliability-HPO of the convex optimizers across problem types and settings. We use the following acronyms: QP (Quadratic Programming), SOCP (Second-Order Cone Programming), SPO (Single-Period Optimization), and MPO (Multi-Period Optimization).

|            | MOSEK | OSQP  | SCS   | ECOS |
|------------|-------|-------|-------|------|
| QP - SPO   | 0.933 | 0.525 | 0.990 | 1.00 |
| QP - MPO   | 0.919 | 0.448 | 0.999 | 1.00 |
| SOCP - SPO | 0.931 | N/A   | 0.929 | 1.00 |
| SOCP - MPO | 0.975 | N/A   | 0.796 | 1.00 |

mance. On the other hand, multi-period optimization notably hinders the performance of all optimizers. All optimizers solve the 503-dimensional single-period optimization problems much faster than the 375-dimensional multi-period optimization problem. We suspect CVXPY causes this performance loss when translating the multi-period optimization problems into the solver-specific forms. This performance loss is evident in ECOS's benchmark. We provide it in Table 1.

ECOS has nearly identical performance in quadratic programming and second-order cone programming for all dimensionalities. On the other hand, multi-period optimization heavily hinders its performance. This shows the performance loss through CVXPY.

## 4.2 RELIABILITY-HPO

Table 2 provides the optimizers' Reliability-HPO. ECOS has no hyperparameters. Hence, it has the highest Reliability-HPO. Among other optimizers, SCS has the highest Reliability-HPO in quadratic programming. MOSEK follows it. In second-order cone programming, MOSEK has the second-highest Reliability-HPO after ECOS.

Multi-period optimization, compared to single-period optimization, does not have a significant impact on MOSEK. However, it reduces OSQP's reliability by 17% and SCS's reliability by 16% on average across quadratic programming and second-order cone programming. Switching from quadratic programming to second-order cone programming does not affect MOSEK's reliability. In contrast, it reduces SCS's reliability by 32% on average across both single-period and multi-period optimization. Overall, SCS exhibits the lowest consistency in reliability values across different problem types and settings.

## 5 CONCLUSION

We address benchmarking algorithms with different hyperparameters. We propose criteria to measure the performance and reliability of such algorithms based on hyperparameter optimization. The Performance-HPO trajectory shows how an algorithm performs based on amounts of hyperparameter optimization budgets. With this trajectory, users are able to identify the most suitable algorithm for their needs. From a reliability standpoint, an algorithm may need careful hyperparameter selection for a successful run. The Reliability-HPO criterion explores this by estimating the expected success rate of the algorithm with random hyperparameters. These two criteria offer a comprehensive framework to compare algorithms with different hyperparameters. This procedure is without bias or human intervention. It applies to any algorithm regardless of its area. We utilize this framework to benchmark convex optimizers through CVXPY. In real-world problems, the SCS solver achieves the best combination of Performance-HPO and Reliability-HPO.

**Limitations and Future Work.** For many deep learning algorithms, GPU acceleration is essential. While the computation of the Performance-HPO trajectory and Reliability-HPO is easily parallelizable, it still demands potentially costly GPU time. Variance reduction techniques in Monte Carlo simulations, such as antithetic sampling (Rubinstein & Kroese, 2016) and sequential Bayesian quadrature (Rasmussen & Ghahramani, 2003), potentially achieve the same accuracy with a lower number of samples.

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

# A  CONVEX OPTIMIZERS

## A.1  OSQP

`OSQP` (Operator Splitting Quadratic Program) is an optimization solver designed to solve large-scale quadratic programs. Quadratic programs involve optimizing a quadratic objective function subject to linear constraints. `OSQP` is particularly efficient for problems with sparse data structures and is able to handle a wide range of problems, including convex and some non-convex ones.

The general form of a quadratic program is:

$$\min_{x} \quad \frac{1}{2}x^T P x + q^T x$$
$$\text{subject to} \quad l \le Ax \le u,$$

where $x \in \mathbb{R}^n$ is the variable to optimize, $P \in \mathbb{R}^{n \times n}$ is a symmetric positive semidefinite matrix, $q \in \mathbb{R}^n$ is a vector, $A \in \mathbb{R}^{m \times n}$ is a constraint matrix, and $l, u \in \mathbb{R}^m$ are vectors defining the lower and upper bounds for the constraints.

The original quadratic program is reformulated using auxiliary variables to separate the objective and constraints:

$$\min_{x,z} \quad \frac{1}{2}x^T P x + q^T x$$
$$\text{subject to} \quad l \le z \le u,$$
$$z = Ax.$$

`OSQP` uses the augmented Lagrangian method, introducing Lagrange multipliers $y$ and a penalty parameter $\rho$:

$$L_\rho(x, z, y) = \frac{1}{2}x^T P x + q^T x + y^T(Ax - z) + \frac{\rho}{2}\|Ax - z\|_2^2.$$

Alternating Direction Method of Multipliers (ADMM) solves the augmented Lagrangian iteratively through three main steps:

- $x$-Update: Solve for $x$ given the current $z$ and $y$:
$$x^{k+1} = \arg\min_{x} \left( \frac{1}{2}x^T P x + q^T x + \frac{\rho}{2}\|Ax - z^k + \frac{y^k}{\rho}\|_2^2 \right).$$

- $z$-Update: Solve for $z$ given the updated $x$ and current $y$:
$$z^{k+1} = \arg\min_{z} \left( \frac{\rho}{2}\|Ax^{k+1} - z + \frac{y^k}{\rho}\|_2^2 \right) \quad \text{subject to} \quad l \le z \le u.$$

- $y$-Update: Update the Lagrange multipliers:
$$y^{k+1} = y^k + \rho(Ax^{k+1} - z^{k+1}).$$

`OSQP` includes several optimization techniques to improve efficiency and robustness:

- Sparse Linear Algebra: Utilizes sparse matrix operations to exploit the sparsity in $P$ and $A$, improving computational efficiency.

- Adaptive Parameter Selection: Adjusts the penalty parameter $\rho$ dynamically to improve convergence rates.

- Polishing: After finding an approximate solution, `OSQP` is able to refine it to achieve higher precision.

## A.2  SCS

`SCS` (Splitting Conic Solver) is an optimization solver designed to solve large-scale convex cone programs. It employs operator splitting techniques to handle various types of convex constraints effectively, making it particularly useful for solving large, sparse optimization problems.

The general form of a cone program is:

$$\min_{x} \quad c^T x$$
$$\text{subject to} \quad Ax + s = b,$$
$$s \in \mathcal{K},$$

.

where $x \in \mathbb{R}^n$ is the variable to optimize, $c \in \mathbb{R}^n$ is a vector, $A \in \mathbb{R}^{m \times n}$ is a constraint matrix, $b \in \mathbb{R}^m$ is a vector, and $\mathcal{K}$ is a convex cone (e.g., non-negative orthant, second-order cone, or positive semidefinite cone).

The optimization problem is reformulated using the primal-dual formulation and then solved iteratively using the Alternating Direction Method of Multipliers (ADMM). The primal-dual formulation is:

$$\min_{x,s} \quad c^T x$$
$$\text{subject to} \quad Ax + s = b,$$
$$s \in \mathcal{K}.$$

SCS uses the ADMM method to decompose the problem into smaller, more manageable subproblems. The key steps in the ADMM algorithm for SCS are:

- $x$-Update: Solve for $x$ given the current $s$ and dual variable $y$:
$$x^{k+1} = \arg\min_{x} \left( c^T x + \frac{\rho}{2} \|Ax + s^k - b + \frac{y^k}{\rho}\|_2^2 \right).$$

- $s$-Update: Solve for $s$ given the updated $x$ and current $y$:
$$s^{k+1} = \arg\min_{s} \left( \frac{\rho}{2} \|Ax^{k+1} + s - b + \frac{y^k}{\rho}\|_2^2 \right) \quad \text{subject to} \quad s \in \mathcal{K}.$$

- $y$-Update: Update the dual variable $y$:
$$y^{k+1} = y^k + \rho(Ax^{k+1} + s^{k+1} - b).$$

SCS includes several techniques to improve efficiency and robustness:

- Preconditioning: Uses preconditioning to improve the problem's conditioning, potentially leading to faster convergence.
- Sparse Linear Algebra: Employs efficient sparse matrix operations to exploit the sparsity in $A$, reducing computational complexity.

### A.3  MOSEK

MOSEK is an optimization solver designed to solve large-scale linear, quadratic, and conic optimization problems, including quadratic and second-order cone programs. It is particularly efficient for high-dimensional problems and is able to handle various problem types, including convex and some non-convex ones.

- Quadratic programs involve optimizing a quadratic objective function subject to linear constraints. The general form of a quadratic program is:
$$\min_{x} \quad \frac{1}{2}x^T P x + q^T x$$
$$\text{subject to} \quad Gx \leq h,$$
$$Ax = b,$$
where $x \in \mathbb{R}^n$ is the optimization variable, $P \in \mathbb{R}^{n \times n}$ is a symmetric positive semidefinite matrix, $q \in \mathbb{R}^n$ is a vector, $G \in \mathbb{R}^{m \times n}$ is a constraint matrix, $h \in \mathbb{R}^m$ is a vector of inequality bounds, $A \in \mathbb{R}^{p \times n}$ is an equality constraint matrix, and $b \in \mathbb{R}^p$ is a vector of equality bounds.

MOSEK uses interior-point methods to solve quadratic programs by:
  - Formulating the Karush-Kuhn-Tucker conditions: These conditions combine the primal and dual formulations into a system of equations.
  - Using a barrier function: This function is introduced to prevent the iterates from leaving the feasible region.

- Applying Newton's method: Newton's method is used to solve the Karush-Kuhn-Tucker conditions iteratively.
  - Updating the barrier parameter: The barrier parameter is gradually reduced to zero, guiding the solution toward optimality.
- Second-order cone programs involve optimizing a linear objective function subject to second-order (or quadratic) cone constraints. The general form of a second-order cone program is:

$$\min_x \quad c^T x$$
$$\text{subject to} \quad \|A_i x + b_i\|_2 \leq c_i^T x + d_i, \quad i = 1, \ldots, m,$$
$$F x = g,$$

where $x \in \mathbb{R}^n$ is the optimization variable, $c \in \mathbb{R}^n$ is the objective function vector, $A_i \in \mathbb{R}^{k_i \times n}$, $b_i \in \mathbb{R}^{k_i}$, $c_i \in \mathbb{R}^n$, and $d_i \in \mathbb{R}$ define the second-order cone constraints, $F \in \mathbb{R}^{p \times n}$ is the equality constraint matrix, and $g \in \mathbb{R}^p$ is the equality constraint vector.

MOSEK uses interior-point methods to solve second-order cone programs by:
- Formulating the primal and dual problems: The primal and dual problems are formulated along with their Karush-Kuhn-Tucker conditions.
- Using a barrier function for cones: A logarithmic barrier function specific to second-order cones is used to ensure that iterates remain within the feasible region of the cones.
- Applying Newton's method for conic problems: Newton's method is applied to solve the perturbed Karush-Kuhn-Tucker conditions iteratively, considering the conic constraints.
- Updating the barrier parameter: Similar to quadratic programs, the barrier parameter is reduced to approach the optimal solution gradually.

MOSEK includes several optimization techniques to improve efficiency and robustness:
- Sparse linear algebra: Utilizes sparse matrix operations to exploit the sparsity in problem data, improving computational efficiency. This is crucial for handling large-scale problems efficiently.
- Presolving: Simplifies the problem before the main optimization phase, reducing problem size and complexity. This involves techniques such as removing redundant constraints and variables.
- Numerical stability: Implements strategies to maintain numerical stability and robustness during optimization. This ensures accurate solutions even for ill-conditioned problems.
- Parallel computation: Uses parallel algorithms to speed up optimization, taking advantage of multi-core processors. This allows MOSEK to handle large-scale problems more efficiently.

## A.4 ECOS

ECOS (Embedded Conic Solver) is an optimization solver designed to solve large-scale convex cone programs. It is particularly efficient for solving second-order cone programs (SOCPs) and linear programs (LPs). ECOS is well-suited for embedded applications due to its low memory footprint and computational efficiency.

The general form of a conic program solved by ECOS is:

$$\min_x \quad c^T x$$
$$\text{subject to} \quad G x + s = h,$$
$$s \in \mathcal{K},$$
$$A x = b,$$

where $x \in \mathbb{R}^n$ is the variable to optimize, $c \in \mathbb{R}^n$ is a vector, $G \in \mathbb{R}^{m \times n}$ and $A \in \mathbb{R}^{p \times n}$ are constraint matrices, $h \in \mathbb{R}^m$ and $b \in \mathbb{R}^p$ are vectors, and $\mathcal{K}$ is a convex cone (e.g., non-negative orthant, second-order cone, or positive semidefinite cone).

ECOS uses an interior-point method to solve the conic program. The interior-point method iteratively improves the solution by moving through the interior of the feasible region defined by the constraints.

The key steps in the interior-point method used by ECOS are:

- Formulate the KKT (Karush-Kuhn-Tucker) conditions for the conic program, which consist of primal feasibility, dual feasibility, and complementary slackness conditions.

- Linearize the KKT conditions around the current iterate and solve the resulting linear system to obtain a search direction.

- Perform a line search to determine the step size along the search direction that ensures progress toward optimality while maintaining feasibility.

- Update the iterate and check for convergence based on predefined criteria, such as the residual norms of the KKT conditions.

`ECOS` includes several optimization techniques to improve efficiency and robustness:

- Preconditioning: Applies preconditioning techniques to improve the conditioning of the linear system, which enhances numerical stability and convergence speed.

- Sparse Linear Algebra: Utilizes sparse matrix operations to exploit the sparsity in the constraint matrices $G$ and $A$, reducing computational complexity and memory usage.

- Inexact Search Directions: Allows for inexact computation of search directions, which is able to accelerate convergence for large-scale problems.

## B  OPTIMIZERS' HYPERPARAMETERS

This section provides the list of tunable hyperparameters for each convex optimizer.

### B.1  OSQP

- `Rho`: This is the step size for the ADMM (Alternating Direction Method of Multipliers) algorithm used by `OSQP`. It controls the rate at which the algorithm progresses towards convergence. The choice of `Rho` potentially affects both the convergence speed and the algorithm's stability. A very small value might lead to slow convergence, while a very large value might cause instability or oscillation.

- `Alpha`: Known as the over-relaxation parameter, `Alpha` is used to improve convergence in the ADMM algorithm. It adjusts the iterative process's trajectory and potentially helps accelerate convergence.

- `Sigma`: This parameter adds a regularization term to the quadratic programming problem being solved by `OSQP`. `Sigma` helps ensure numerical stability and prevents the algorithm from taking too large steps. It effectively adds a small positive value to the diagonal of the quadratic term in the objective function, which potentially helps in situations where the problem might be ill-conditioned or near-degenerate.

### B.2  SCS

- `Alpha`: This parameter is used in the over-relaxation step within the ADMM (Alternating Direction Method of Multipliers) framework of `SCS`. `Alpha` influences the convergence rate by modifying the trajectory of the algorithm's iterations.

- `Rho_x`: This hyperparameter controls the x-update step size in the ADMM algorithm. Adjusting `Rho_x` affects the algorithm's stability and convergence speed. A suitable value helps balance the progress rate towards an optimal solution and the overall stability of the algorithm.

- `Scale`: `Scale` is a scaling factor applied to the data of the problem before solving it. This hyperparameter helps adjust the problem's condition number, making it more suitable for numerical solving. Proper scaling potentially significantly enhances solver performance and stability.

- `Acceleration_Lookback`: This parameter sets the number of previous iterations that `SCS` will consider when using acceleration techniques like Anderson acceleration. A higher `Acceleration_Lookback` value allows the solver to potentially utilize more historical data to speed up convergence but at the cost of increased memory usage and computational overhead.

- `Acceleration_Interval`: `Acceleratio_interval` specifies how frequently the acceleration techniques are applied. A lower value means more frequent application, which could speed up convergence but increase computational overhead.

### B.3  MOSEK

- `Msk_Dpar_Intpnt_Tol_Rel_Step`: This hyperparameter sets the relative step size tolerance in the interior-point method. It controls the magnitude of the steps taken towards the optimal solution relative to the current position. A smaller value results in smaller and more precise steps, potentially leading to a more accurate solution, but may increase the number of iterations and computation time.

- `Msk_Dpar_Intpnt_Tol_Psafe`: This parameter is related to the primal safety margin in the interior-point method. It sets a tolerance level for the proximity to the boundaries of the feasible region in the primal space. Increasing this value ensures a larger safety margin, thereby reducing the risk of numerical issues, but could impact the optimality and convergence speed of the solution.

- `Msk_Dpar_Intpnt_Tol_Dsafe`: This parameter determines the dual safety margin in the interior-point method. It controls the safety margin in the dual space, thus balancing the numerical stability with the quality and convergence rate of the obtained solution.

- `Msk_Dpar_Intpnt_Tol_Path`: Finally, this hyperparameter sets the tolerance for adhering to the central path in the interior-point method. The central path represents the ideal trajectory towards the optimal solution. A lower hyperparameter value means the algorithm will more closely follow this path, potentially increasing accuracy at the cost of greater computational effort.

### B.4  ECOS

This solver has no hyperparameters.

## C  HYPERPARAMETER OPTIMIZERS

- DiRect (Gablonsky & Kelley, 2001): The global derivative-free optimization algorithm `DiRect` (Dividing Rectangles) systematically explores and divides the search space to find the global minimum. It is particularly effective for problems where gradient information is unavailable or unreliable. `DiRect`'s comprehensive search strategy makes it robust against local minima but is potentially computationally intensive.

- DiRect_L (Gablonsky & Kelley, 2001): This localized version of `DiRect` focuses more intensively on promising regions of the search space, enhancing efficiency in situations where the global minimum is suspected to be in a particular area. While it offers faster convergence than the standard `DiRect`, it may overlook the global minimum if it lies outside the focused regions.

- CRS2_LM (Kaelo & Ali, 2006a): The Controlled Random Search algorithm with Local Mutation (`CRS2_LM`) is a stochastic method suitable for complex global optimization problems, particularly those with multiple local minima. It combines random search techniques with local mutation strategies to explore the search space, balancing exploration and exploitation.

- ESCH (Kaelo & Ali, 2006b): The `ESCH` optimizer, based on an Evolution Strategy by Cholesky Hessian, is an evolutionary algorithm designed for complex, multimodal global optimization problems. It employs mutation and selection strategies inspired by natural evolutionary processes, making it effective in diverse problem landscapes.

- ISRES (Kaelo & Ali, 2006c): The Improved Stochastic Ranking Evolution Strategy (`ISRES`) is an evolutionary algorithm focusing on constraint handling. It is particularly adept at tackling global optimization problems involving nonlinear constraints, using a ranking-based mechanism to guide the search process.

- COByLA (Powell, 1994a): The Constrained Optimization By Linear Approximations (`COByLA`) algorithm is a local, derivative-free method for constrained optimization. It uses linear approximations to model the objective function and constraints, making it suitable for problems lacking gradient information.

- BOByQA (Powell, 1994a): `BOByQA` (Bounded Optimization By Quadratic Approximations) is designed for derivative-free local optimization, particularly in problems with bound constraints. It uses quadratic models to approximate the objective function, facilitating efficient convergence to a local minimum.

- NEWUOA_Bound (Powell, 2006): An extension of the `NEWUOA` algorithm, `NEWUOA_Bound` is tailored for local optimization problems with bound constraints. It is effective when derivative information is absent, using a trust-region approach to guide the optimization.

- PrAxis (Brent, 1969): The `PrAxis` (Principal Axis) algorithm is a local optimization method focusing on derivative-free unconstrained problems. It employs principal axis methods for optimization, making it suitable for smooth, well-behaved functions.

- Nelder_Mead (Nelder & Mead, 1965): `Nelder_Mead` is a heuristic method for multidimensional unconstrained optimization without derivatives. It adapts well to non-smooth functions and is known for its simplicity and broad applicability.

- Sbplx (Powell, 1994b): This variant of the `Nelder-Mead` algorithm is adapted for problems with bound constraints. The `Sbplx` method maintains the general approach of the simplex algorithm but includes modifications to handle the constraints, making it suitable for local optimization in a range of applications.

