# OpenReview forum: "A Comprehensive Framework for Benchmarking Algorithms Across Hyperparameter Spaces"
_ICLR.cc/2025/Conference — ICLR 2025 Conference Withdrawn Submission_

### Official Review · Reviewer_gp1X · 2024-10-31

**Soundness:** 2
**Presentation:** 1
**Contribution:** 1
**Rating:** 1
**Confidence:** 4

**Summary:**

The paper proposes a general framework for benchmarking algorithms that have hyperparameters.
This framework is composed of two key criteria for evaluation:
Performance-HPO trajectories and the Reliability-HPO criterion.
The Performance-HPO trajectory is in essence the standard anytime performance visualization of an algorithm given differently tried hyperparameter configurations over time.
The Reliability-HPO criterion is designed to capture the expected value of an algorithm's success rate across
hyperparameters.
The paper demonstrates practical usefulness of these two criteria conducting a benchmark of widely used convex optimization algorithms across various numerical optimization problem types, settings and dimensionalities, i.e.
benchmarking multiple convex optimizers from the CVXPY library using the proposed framework across quadratic programming and second-order cone programming problems in single- and multi-period settings.
The framework tries to establish unbiased and reproducible benchmarks for hyperparameter sensitive algorithms.

**Strengths:**

The paper is generally written clearly and the problem of algorithms being sensitive to hyperparameters is relevant.
The framework is in general applicable to any setting where algorithms have hyperparameters that are optimized.
In principle, significance to the ML and DL community is given as models are sensitive to their hyperparameters or architectural choices in general.

**Weaknesses:**

Originality is strongly limited.
In general the statement that algorithms dependency on hyperparameters can be crucial is correct and does apply to the field of ML and DL as demonstrated by the sub-fields of Hyperparameter Optimization (HPO) and Neural Architecture Search (NAS) to which the authors try to make a connection in the introduction (via the bi-level optimization problem).
However, the ideas of the two criteria are generally not novel but have been used in the fields of HPO, NAS and AutoML (see e.g., the book on Automated Machine Learning [1]) to large extent already (especially looking at the Performance-HPO criterion; maybe less for the Reliability-HPO criterion which I will discuss below).

Performance-HPO is simply the anytime visualization of the cumulative performance of an algorithm over time (where time is the number of different configurations tried - with the exception that not only a single optimizer may be used to generate the different configurations, but multiple ones in parallel).
This general idea how to visualize performance, is however already a well-established standard criterion in the field of HPO, NAS and AutoML.
See e.g., existing benchmarking libraries such as HPOBench [2], HPO-B [3] or YAHPO Gym [4] and how they visualize anytime performance of HPO algorithms for a given model or on average (e.g., Figure 8 in the Appendix of [2]).
If we say that an optimizer is now a conglomerate of optimizers and we visualize the anytime performance of this conglomerate, then there appears to be no novelty over existing visualization techniques.
These visualization also already have made their way to general benchmarking studies of ML models that use HPO, see e.g., [5]

Reliablity-HPO on the other hand tries to measure the probability of an algorithm's success over random hyperparameter configurations. Success is defined as exceeding a certain performance threshold within a time limit.
For HPO and NAS of ML and DL models, the estimated generalization error depending on hyperparameters is generally a black-box so it is difficult to come up with a reasonable threshold to define success a priori without any knowledge.
For numerical optimization problems this criterion might be applicable where on test functions the optimum value is usually known so that epsilon thresholds around its optimum may be reasonable.
In the context of ML and DL models, however, this is not straightforward and limits practical applicability.

In general, the paper does not clearly define its scope and there is a gap between problem statement and the experimental investigations.
On the one hand side, it is concerned with HPO of ML and DL models and introduces the bi-level optimization problem and discusses it in the introduction.
On the other hand, empirical evaluation of the framework is performed on numerical optimization algorithms suitable to solve convex optimization problems outside of the DL and ML scope (i.e., the solvers would correspond to the ML models and their parameters would correspond to the hyperparameters of the ML models).
This feels like a disconnection between the point the paper originally tries to make -- tailored to the DL and ML community -- and what is presented based on their benchmark study.

The paper presents many large multi-facet plots in the main section that take up a lot of space and provide comparably little information (e.g. anytime performance of only two distinct algorithms as in Figure 2; Figure 3 carries more information; but in total there are two full pages of multi-facet figures in the main section that are not easy to visually digest).
Maybe these Figures could be refactored and combined which would free up space in the main section which could be used for some more technical depth on the criteria and/or additional actual HPO/NAS benchmark experiments.

[1] https://link.springer.com/book/10.1007/978-3-030-05318-5
[2] https://arxiv.org/pdf/2109.06716
[3] https://arxiv.org/abs/2106.06257
[4] https://arxiv.org/abs/2109.03670
[5] https://arxiv.org/abs/2207.08815

**Questions:**

Eq (3) somewhat lacks clarity in notation as the end result should be a curve over time because in its current formulation (3) returns the best value for a given time point $t$ with $t \in [0, T]$ without explicitly collecting these over time as a tuple of values.
Maybe the authors could improve notation here?

Have you considered running actual HPO or NAS experiments of ML and DL Models?
How do you suggest addressing the computational expense in Reliability-HPO when working with high-dimensional hyperparameter spaces where real evaluations can be very costly?
Also, in real ML and DL settings it might be prohibitively expensive to run many different HPO optimizers for a given model in parallel for many evaluations which would be your proposed way how to construct the Performance-HPO trajectory.
Do you have any suggestions how to make your approach applicable in these settings?

**Details Of Ethics Concerns:**

There are no ethical issues.

---

### Official Review · Reviewer_Z9ok · 2024-11-01

**Soundness:** 2
**Presentation:** 3
**Contribution:** 2
**Rating:** 1
**Confidence:** 2

**Summary:**

In addition to the effect of hyperparameter optimization, the author designs two criteria to facilitate the construction of algorithm comparison benchmarks.

**Strengths:**

The author has come up with two criteria to help evaluate the merits and demerits of different algorithms fairly, so that their hyperparameter tuning does not affect the judgment of the algorithm.

**Weaknesses:**

1. The framework requires that the total budget be specified in advance. The optimizer of the multi-fidelity type is invalid. Multi-fidelity methods like BOHB, Hyperband, will change their budget allocation in each epoch according to the pre-given total budget.

2. In actual use, to select the best algorithm for the current task, it is impossible to adjust all optimizers to find the best hyperparameter. The guidance function of the benchmark result for the actual task is still unclear.

3. The reliability criterion design is also unreasonable. If the given precision threshold varies greatly, the reliability of an algorithm without hyperparameters will jump between 0 and 1. Moreover, it is unreasonable to extract hyperparameters from log-uniforms. The hyperparticipation algorithm itself is based on the matching relationship. A large amount of hyperparameter space is not used in practice. These parts should not be involved in reliability measurement.

4. The HPO algorithms listed in Appendix C are old algorithms from 20 years ago.

**Questions:**

See above

---

### Official Review · Reviewer_Dsvu · 2024-11-03

**Soundness:** 1
**Presentation:** 1
**Contribution:** 1
**Rating:** 1
**Confidence:** 5

**Summary:**

The paper introduces a framework to address the challenges of evaluating algorithms whose performance varies with HPO. The framework aims to create fair, scalable comparisons of algorithms by defining two primary criteria: the "Performance-HPO trajectory" and "Reliability-HPO." The Performance-HPO trajectory is a metric that records how algorithm performance changes relative to different HPO budget allocations to provide insights into the efficiency of algorithms as the HPO budget increases. Meanwhile, the Reliability-HPO metric evaluates an algorithm's likelihood of achieving successful outcomes across various hyperparameter settings. This reliability measure is based on Monte Carlo simulations to provide a probabilistic view of an algorithm’s robustness. The authors validate their framework by applying it to standard convex optimization algorithms, solving problems of different types, including quadratic and second-order cone programming in both single- and multi-period optimization settings.

**Strengths:**

The authors recognize the need for a structured, standardized approach to benchmarking algorithms. The dual approach of introducing Performance-HPO and Reliability-HPO enables a more comprehensive evaluation that considers both an algorithm's efficiency in using optimization budgets and its consistency across diverse hyperparameter settings. By balancing performance against reliability across hyperparameter settings, the framework allows for a thorough, unbiased assessment.

Furthermore, by designing the framework to accommodate various algorithm types and domains, the paper addresses a need for reproducibility and fairness in empirical comparisons.

**Weaknesses:**

The paper aims to discuss hyperparameter optimization, but instead focuses on tuning parameters of solvers for optimization problems, effectively making it an algorithm configuration task rather than HPO. See Schede, Elias, et al. "A survey of methods for automated algorithm configuration." (JAIR'22) for discussing the difference.

The references for HPO are outdated, primarily mentioning random search, which is not a state-of-the-art HPO method. The paper only briefly touches on established HPO approaches like Bayesian optimization, but overlooks widely used modern frameworks like Optuna (Akiba et al., 2019) and SMAC (Lindauer et al., 2023). Similarly, the authors briefly mention evolutionary algorithms but fail to mention relevant frameworks, e.g., DEHB (Awad et al., 2021).

Generally, the listed references are insufficient for contextualizing the work within recent advances in HPO, limiting its relevance to current literature, including e.g. Bischl, Bernd, et al. "Hyperparameter optimization: Foundations, algorithms, best practices, and open challenges." (2023). The authors fail to place their work in the context of tuning convex optimizers as well.

While the title claims to introduce a "comprehensive framework," the paper essentially evaluates four convex optimizers on toy problems involving different stock sample sizes, resulting in limited scope. The generalizability of the approach has not been investigated. Overall, the contribution of this paper is not clear to me.

The study lacks reproducibility, as the authors do not clearly outline the specifics of the optimization, configuration, or sampling processes.
It seems that the focus of the work is more aligned with numerical analysis than with machine learning, making it a less appropriate fit for the intended scope and the ICLR conference.

Lastly, the writing style lacks clarity, with a tendency toward repetitive statements rather than a coherent, structured flow of ideas. The analyzed parameters are only vaguely explained in Appendix B. A proper explanation is important for assessing the validity of the results and ensuring their informed interpretation.

**Questions:**

* How exactly do the convex optimizers interact with the algorithms chosen for parameter tuning?
* How does the performance evaluation of the optimizers instantiated with different parameters guarantee generalizability? How does this align with the goal of HPO and benchmarking algorithms across hyperparameter spaces?
* Do the results generalize to other datasets/problems?
* Could the authors provide a more explicit (formal) definition of the objective function/the problem design?
* What are the current references for tuning convex optimizers?
* What insights does the reader get from analyzing the ECOS optimizer that does not have any parameters to tune?
* How does the reliability analysis relate to HPO? What insights about the optimization process and its results do we get from the reliability analysis?
* How robust is the definition of reliability (p. 3 & 6)? Is it implied that an optimizer without hyperparameters could be beneficial due to maximum reliability? Reliability should not imply no variance in performance, but rather good performance given s

---

### Official Review · Reviewer_V1tx · 2024-11-04

**Soundness:** 2
**Presentation:** 2
**Contribution:** 2
**Rating:** 3
**Confidence:** 2

**Summary:**

The paper presents a general framework to benchmark algorithms based on their different hyper-parameters performances.
To decide, they propose two metrics to trade off, Performance HPO trajectory and Reliability HPO.
Performance HPO trajectory is a plot that shows how an algorithm's results vary if allocating more budget for HPO. Having such a plot for different algorithms, the user decides which one is more suitable, on different level budgets.
Reliability HPO is the expected value of the success rate of an algorithm, across hyperparameters using Monte Carlo simulation.

Given a problem, and some algorithm candidates, the framework consists of getting these two metrics. First, they run different off the shelf HPO methods, and for each budget, they pick the method that gives the best result at each timestep. By integrating over all the budgets, we can get a performance HPO trajectory for each algorithm.
Then, after defining a success condition for the problem, they sample a large number of hyperparameters. They obtain the Reliability HPO by computing the average success rate over these hyperparameters.

These two metrics give the user the ability to choose between algorithms based on their budget (Performance HPO trajectory) and also how sensitive an algorithm is to hyperparameters tuning (Reliability HPO).

To test the method, the authors created optimization problems and tested different convex optimizers as a benchmark.

Overall, I have trouble understanding the goal of the paper. I don’t really understand what to do with the methodology, and how it compares to existing methodology in the field of global optimization, see questions. Therefore, I cannot recommend acceptance.

**Strengths:**

The framework idea is simple, yet effective to benchmark between different algorithms and to give a feeling of which algorithms are more suitable for a problem and the budget and time needed to do HPO. It is also very useful for democratizing these methods across different disciplines.

**Weaknesses:**

* The framework is easily generalizable across different problems, algorithms and HPO methods. But given a totally new problem, the user needs to run all the experiments (different and expensive HPO methods for all the budgets, and compute the Monte Carlo simulations). These loops can be overwhelming for some kinds of problems. Nevertheless, if you have an old plot of Performance HPO trajectory, the user can already have a feeling on how much budget to invest and on which kind of algorithms.
* The success rate definition is a tricky hyper parameter but very important for the framework
* The paper is applied to portfolio optimization, and would have been interesting to test it on HPO benchmarks. Especially as the authors discuss hyperparameter optimization, which is commonly associated with machine learning. One suggestion is to test “off-the-shelf” algorithms such as random forest or catboost against something that requires more hyperparameter tuning.
* Related work could be improved, see the links I gave.

## Minor issues:
* The statement “approximately 45% of hyperparameter optimization [...] is manual (Gundersen et al., 2022)” uses the wrong reference to back up the claims. This was found in an original study by [Xavier Bouthillier and Gaël Varoquaux](https://bouthilx.github.io/publication/2020-01-21-survey-neurips-iclr).
* The paper should cite [Accounting for Variance in Machine Learning benchmarks](https://bouthilx.github.io/publication/2021-04-07-accounting-for-variance).
* Melis et al. (2017) has been published at ICLR, please cite the published version.
* Please cite Bergstra and Bengio (2012) for random search.

**Questions:**

* How is t in Equation 4 related to t in Equation 3?
* I do not understand how the “Performance-HPO” can be used in practice. Because it is the best performance among multiple hyperparameter optimization algorithms, how could I achieve this performance in practice?
* I do not understand how the “Performance-HPO” can be used in practice. If the hyperparameter optimization algorithm depends on the maximal budget t, we would have to run one hyperparameter optimization run for each t \in [0, T], right?
* I do not understand how the “Performance-HPO” can be used in practice. After obtaining the curve, we have solved the problem, but we cannot make any statement about similar problems. Is this correct?
* How do the proposed measures relate to measures used to evaluate continuous optimization algorithms? See for example [this presentation](http://www.cmap.polytechnique.fr/~nikolaus.hansen/cocotenyears-leiden2020.pdf) and [Benchmarking derivative-free optimization algorithms](https://epubs.siam.org/doi/10.1137/080724083)

---

### Note · Authors · 2024-11-21

I have read and agree with the venue's withdrawal policy on behalf of myself and my co-authors.